# Thinking like a machine — generating visual rationales through latent space optimization

## Abstract

Interpretability and small labelled datasets are key issues in the practical application of deep learning, particularly in areas such as medicine. In this paper, we present a semi-supervised technique that addresses both these issues simultaneously. We learn dense representations from large unlabelled image datasets, then use those representations to both learn classifiers from small labeled sets and generate visual rationales explaining the predictions.

Using chest radiography diagnosis as a motivating application, we show our method has good generalization ability by learning to represent our chest radiography dataset while training a classifier on an separate set from a different institution. Our method identifies heart failure and other thoracic diseases. For each prediction, we generate visual rationales for positive classifications by optimizing a latent representation to minimize the probability of disease while constrained by a similarity measure in image space. Decoding the resultant latent representation produces an image without apparent disease. The difference between the original and the altered image forms an interpretable visual rationale for the algorithm's prediction. Our method simultaneously produces visual rationales that compare favourably to previous techniques and a classifier that outperforms the current state-of-the-art.

## 1 Introduction

Deep learning as applied to medicine has attracted much interest in recent years as a potential solution to many difficult problems in medicine, such as the recognition of diseases on pathology slides or radiology images. However, adoption of machine learning algorithms in fields such as medicine relies on the end user being able to understand and trust the algorithm, as incorrect implementation and errors may have significant consequences. Hence, there has recently been much interest in interpretability in machine learning as this is a key aspect of implementing machine learning algorithms in practice. We propose a novel method of creating visual rationales to help explain individual predictions and explore a specific application to classifying chest radiographs.

There are several well-known techniques in the literature for generating visual heatmaps. Gradient based methods were first proposed in 2013 described as a saliency map in Simonyan et al. (2013), where the derivative of the final class predictions is computed with respect to the input pixels, generating a map of which pixels are considered important. However, these saliency maps are often unintelligible as convolutional neural networks tend to be sensitive to almost imperceptible changes in pixel intensities, as demonstrated by recent work in adversarial examples. In fact, obtaining the saliency map is often the first step in generating adversarial examples as in Goodfellow et al. (2014). Other recent developments in gradient based methods such as Integrated Gradients from Sundararajan et al. (2017) have introduced fundamental axioms, including the idea of sensitivity which helps focus gradients on relevant features.

Occlusion sensitivity proposed by Zeiler & Fergus (2013) is another method which covers parts of the image with a grey box, mapping the resultant change in prediction. This produces a heatmap where features important to the final prediction are highlighted as they are occluded. Another well-known method of generating visual heatmaps is global average pooling. Using fully convolutional neural networks with a global average pooling layer as described in Zhou et al. (2016), we can

examine the class activation map for the final convolutional output prior to pooling, providing a low resolution heatmap for activations pertinent to that class.

A novel analysis method by Ribeiro et al. (2016) known as locally interpretable model-agnostic explanations (LIME) attempts to explain individual predictions by simulating model predictions in the local neighbourhood around this example. Gradient based methods and occlusion sensitivity can also be viewed in this light — attempting to explain each classification by changing individual input pixels or occluding square areas.

However, sampling the neighbourhood surrounding an example in raw feature space can often be tricky, especially for image data. Image data is extremely complex and high-dimensional — hence real examples are sparsely distributed in pixel space. Sampling randomly in all directions around pixel space is likely to produce non-realistic images.

LIME's solution to this is to use superpixel based algorithms to oversegment images, and to perturb the image by replacing each superpixel by its average value, or a fixed pre-determined value. While this produces more plausible looking images as opposed to occlusion or changing individual pixels, it is still sensitive to the parameters and the type of oversegmentation used — as features larger than a superpixel and differences in global statistics may not be represented in the set of perturbed images. This difficulty in producing high resolution visual rationales using existing techniques motivates our current research.

## 2 METHODS

We introduce a novel method utilizing recent developments in generative adversarial networks (GANs) to generate high resolution visual rationales. We demonstrate the use of this method on a large dataset of frontal chest radiographs by training a classifier to recognize heart failure on chest radiographs, a common task for doctors.

Our method comprises of three main steps — we first use generative adversarial networks to train a generator on an unlabelled dataset. Secondly, we use the trained generator as the decoder section of an autoencoder. This enables us to encode and decode, to and from the latent space while still producing high resolution images. Lastly, we train simple supervised classifiers on the encoded representations of a smaller, labelled dataset. We optimize over the latent space surrounding each encoded instance with the objective of changing the instance's predicted class while penalizing differences in the resultant decoded image and the original reconstructed image. This enables us to visualize what that instance would appear as if it belonged in a different class.

Firstly, we use the Wasserstein GAN formulation by Arjovsky et al. (2017) and find that the addition of the gradient penalty term helps to stabilize training as introduced by Gulrajani et al. (2017). Our unlabelled dataset comprises of a set of 98,900 chest radiograph images, which are scaled to 128 by 128 pixels while maintaining their original aspect ratio through letterboxing, and then randomly translated by up to 8 pixels. We use a 100 dimensional latent space. Our discriminator and generator both use the DCGAN architecture while excluding the batch normalization layers and using Scaled Exponential Linear Units described in Klambauer et al. (2017) as activations except for the final layer of the generator which utilized a Tanh layer. We train the critic for 4 steps for each generator training step. The GAN training process was run for 200k generator iterations before visually acceptable generated images were produced. ADAM was used as the optimizer with the generator and discriminator learning rates both set to $5 \times 10^{-5}$.

In the next step, we use the trained generator as the decoder for the autoencoder. We fix the weights of the decoder during training and train our autoencoder to reproduce each of the images from the unlabelled dataset. The unlabelled dataset was split by patient in a 15 to 1 ratio into a training and validation set. We minimize the Laplacian loss between the input and the output, inspired by Bojanowski et al. (2017). Minimal overfitting was observed during the training process even when the autoencoder was trained for over 1000 epochs, as demonstrated in 2.

We then train a classifier on a smaller labelled dataset consisting of 7,391 chest radiograph images paired with a B-type natriuretic peptide (BNP) blood test that is correlated with heart failure. This test is measured in nanograms per litre, and higher readings indicate heart failure. This is a task of real-world medical interest as BNP test readings are not often available immediately and offered at

all laboratories. Furthermore, the reading of chest radiograph images can be complex, as suggested by the widely varying levels of accuracy amongst doctors of different seniority levels reported by Kennedy et al. (2011). We perform a natural logarithm on the actual BNP value and divide the resultant number by 10 to scale these readings to between 0 and 1. This task can be viewed as either a regression or classification task, as a cut-off value is often chosen as a diagnostic threshold. In this paper, we train our network to predict the actual BNP value but evaluate its AUC at the threshold of 100ng/L. We choose AUC at this threshold as this is the cut-off suggested by Lokuge et al. (2009), and AUC is a widely used metric of comparison in the medical literature.

We augment each labelled image and encode it into the latent space using our previously trained autoencoder. To prevent contamination, we separate our images by patient into a training and testing set with a ratio of 4 to 1 prior to augmentation and encoding. We demonstrate the success of simple classifiers upon this latent representation, including a 2 layer multilayer perceptron with 256 hidden units as well as a linear regressor.

To obtain image specific rationales, we optimize over the latent space starting with the latent representation of the given example. We fix the weights of the entire model and apply the ADAM optimizer on a composite objective comprising of the output value of the original predicted class and a linearly weighted mean squared error term between the decoded latent representation and the decoded original representation. We cap the maximum number of iterations at 5000 and set our learning rate at 0.1. We stop the iteration process early if the cutoff value for that class is achieved. The full algorithm is described in Algorithm 1. This generates a latent representation with a different prediction from the initial representation. The difference between the decoded generated representation and the decoded original representation is scaled and overlaid over the original image to create the visual rationale for that image. We use gradient descent to optimize the following objective:

$$z_{\text{target}} = \arg\min_{z} \ L_{\text{target}}(z) + \gamma \|X - G(z)\|^2 \tag{1}$$

$$X_{\text{target}} = G(z_{\text{target}}) \tag{2}$$

Where $X$ is the reconstructed input image (having been passed through the autoencoder); $X_{\text{target}}$ and $z_{\text{target}}$ are the output image and its latent representation. $G$ is our trained generator neural network. $\gamma$ is a coefficient that trades-off the classification and reconstruction objectives. $L_{\text{target}}$ is a target objective which can be a class probability or a regression target. The critical difference between our objective and the one used for adversarial example generation is that optimization is performed in the latent space, not the image space.

---

**Algorithm 1** Visual rationale generation

**Require:** $\alpha$, learning rate
    $\gamma$, image similarity penalty
    $\rho$, cutoff value
**Require:** $x$, the initial input
    $f : x \to z$, a function approximating the mapping between image and latent space
    $g : z \to x$
    $h(z)$, classifier predicting value from $z$
1:  $z_0 \leftarrow z \leftarrow f(x)$
2: **repeat**
3:    $d \leftarrow \langle (g(z) - g(z_0))^2 \rangle$
4:    $y \leftarrow h(z)$
5:    $z \leftarrow z + \alpha * ADAM(z, y + \gamma d)$
6: **until** $y > \rho$
7: **return** $g(z_0) - g(z)$

---

We also apply our method to external datasets and demonstrate good cross-dataset generalization, in particular the National Institutes of Health (NIH) ChestX-ray8 dataset comprising of 108,948 frontal chest radiographs, recently released by Wang et al. (2017). We downsize the provided images to work with our autoencoder and split this by patient into a training, validation and testing set in the 7:1:2 ratio used by the dataset's authors. We encode these images into the latent space and

apply a 6 layer fully connected neural network with 100 hidden units in each layer utilizing residual connections. We train this with a batch size of 2048. This architecture is fully described in figure 1.

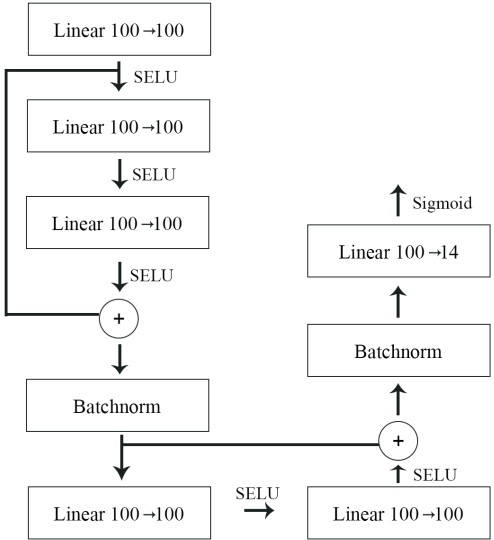

Figure 1: Classifier used for ChestX-ray8 dataset

To evaluate the usefulness of the generated visual rationales, we conduct an experiment where we compare visual rationales generated by a classifier to one which is contaminated. We train the classifier directly on the testing examples and over train until almost perfect accuracy on this set is achieved. We reason that the contaminated classifier will simply memorize the testing examples and hence will not be able to produce useful rationales.

We also apply our method to the well known MNIST dataset and apply a linear classifier with a 10 way softmax. In order to generate our visual rationales we select an initial class and a target class — we have chosen to transform the digit 9 to the digit 4 as these bear physical resemblance. We alter our optimization objective by adding a negatively weighted term for the predicted probability of the target class as described in Algorithm 2.

---
**Algorithm 2** Visual rationale generation for multiclass predictors
---
**Require:** $\alpha$, learning rate
  $\gamma$, image similarity penalty
  $\rho$, cutoff value
  $\beta$, target class weighting
  $t$, target class
**Require:** $x$, the initial input
  $f : x \rightarrow z$, a function approximating the mapping between image and latent space
  $g : z \rightarrow x$
  $h_c(z) \rightarrow P(c|z)$, classifier predicting class probability from $z$
1:  $z_0 \leftarrow z \leftarrow f(x)$
2:  $m \leftarrow \operatorname{argmin}_i h_i(z_0)$
3:  **repeat**
4:  $d \leftarrow \langle (g(z) - g(z_0))^2 \rangle$
5:  $y_m \leftarrow h_m(z)$
6:  $y_t \leftarrow h_t(z)$
7:  $z \leftarrow z + \alpha * ADAM(z, y_m - \beta y_t + \gamma d)$
8:  **until** $y_m > \rho$
9:  **return** $g(z_0) - g(z)$
---

## 3 RESULTS

To illustrate the fidelity of our autoencoder we reconstruct each image in a smaller labelled set which has not been seen during training. The reconstructed images are show in Fig. 3. These images are obtained by simply encoding the input image into the latent representation and subsequently decoding this representation again. MSE loss and the Laplacian loss functions are shown in Fig 2.

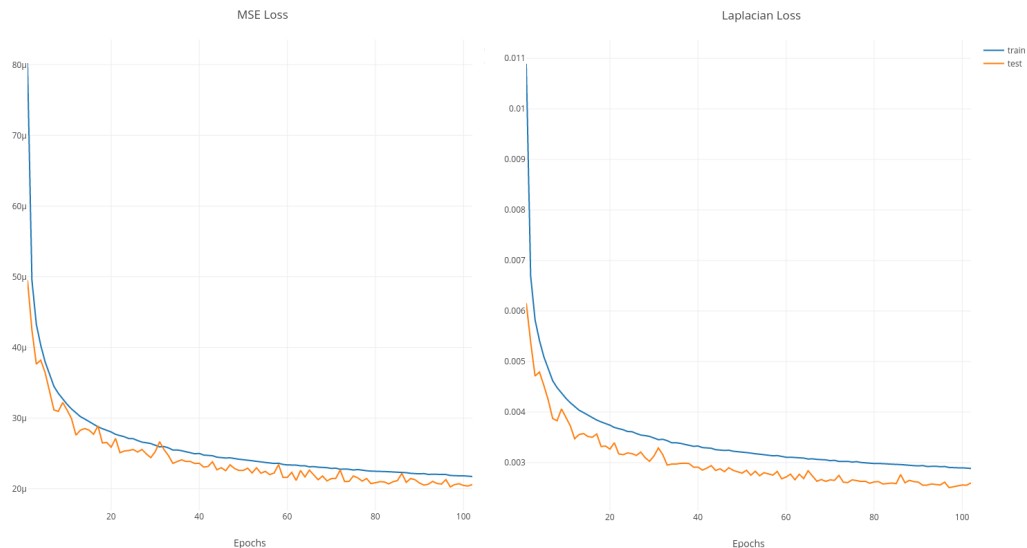

Figure 2: Autoencoder loss function. Left: MSE loss, Right: Laplacian loss

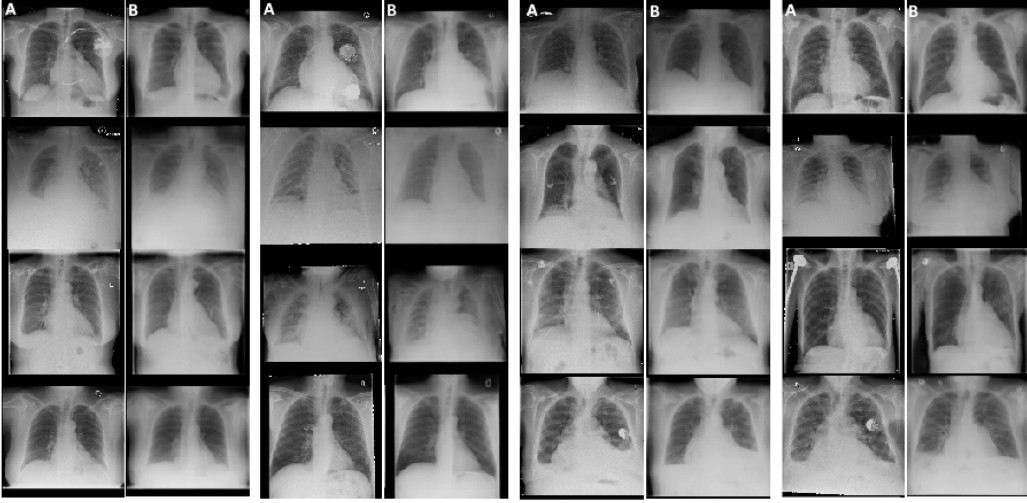

Figure 3: Columns A: Original images. Columns B: reconstructed images

In the heart failure classification task, we threshold the known BNP values at 100ng/L to get binary labels as suggested by Lokuge et al. (2009). Our semi-supervised model achieves an AUC of 0.837 using a linear regressor as our final classifier with an ROC curve as shown in Fig 4. This is comparable to the AUC obtained by a multilayer perceptron.

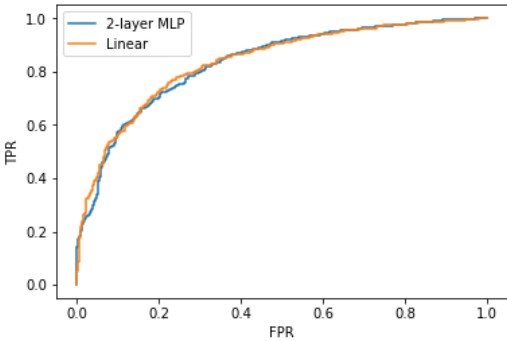

Figure 4: ROC plot for BNP prediction

In Fig 5 we demonstrate an example of the algorithm's reconstruction of a chest radiograph from a patient with heart failure, as well as the visualization of the same patient's chest radiograph without heart failure. We subtract the visualization of the radiograph without heart failure from the original reconstructed image and superimpose this as a heatmap on the original image to demonstrate the visual rationale for this prediction.

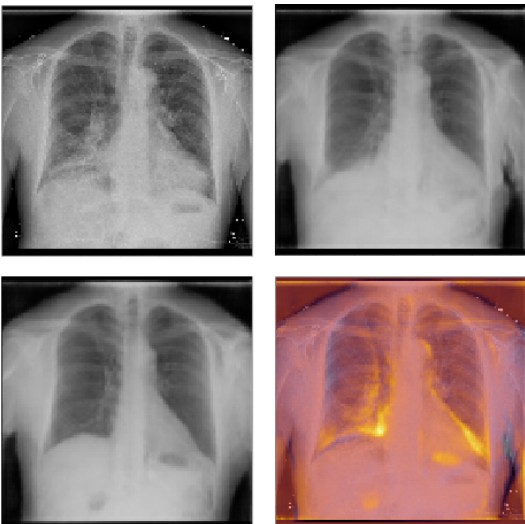

Figure 5: Top left: original image. Top right: reconstructed image. Bottom left: image visualized without heart failure. Bottom right: superimposed visual rationale on original image

For the same image, we apply the saliency map method, integrated gradients, the occlusion sensitivity method with a window size of 8, as well as LIME to obtain Fig. 6 for comparison. All of these methods yield noisy and potentially irrelevant features as compared to our method of generating visual rationales.

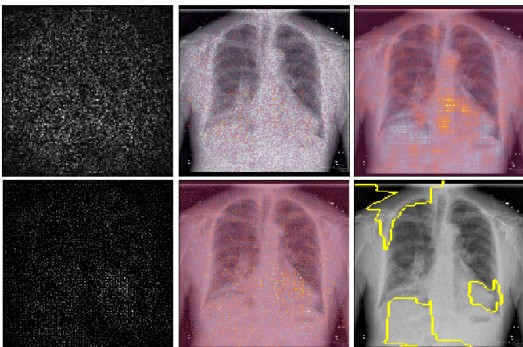

Figure 6: Comparison of other methods - top left to bottom right: Saliency map, saliency map overlaid on original image, heatmap generated via occlusion sensitivity method, Integrated gradients, integrated gradients overlaid on original image, LIME output

We apply our classifier as described above to the chest radiograph dataset released by the NIH recently and achieve results similar to or exceeding that of the baseline results reported in the original dataset. ROC curves are demonstrated in Fig 7. Comparison AUC results are reported in Table 1. We show that even without repeating the autoencoder or GAN training process on the new dataset, we are able to classify encoded representations of these chest radiographs with an accuracy comparable to or exceeding the performance of the published baseline network, which utilizes various state of the art network architectures as well as higher resolution images.

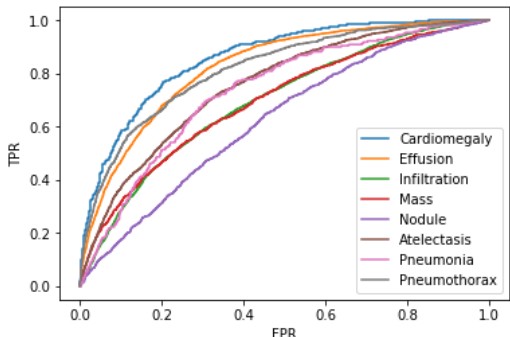

Figure 7: ROC curves for Chest X-Ray8 dataset

|  | Ours | (Wang, 2017) |
|---|---|---|
| Atelectasis | **0.7546** | 0.7069 |
| Cardiomegaly | **0.8589** | 0.8141 |
| Effusion | **0.8243** | 0.7362 |
| Infiltration | **0.6945** | 0.6128 |
| Mass | **0.6958** | 0.5644 |
| Nodule | 0.6247 | **0.7164** |
| Pneumonia | **0.7346** | 0.6333 |
| Pneumothorax | **0.8164** | 0.7891 |

Table 1: Comparison AUC results for ChestX-ray8 dataset

We apply our method to the MNIST dataset and demonstrate class switching between digits from 9 to 4 and 3 to 2. Figure 8. demonstrates the visual rationales for why each digit has been classified as a 9 rather than a 4, as well as the transformed versions of each digit. As expected, the top horizontal line in the digit 9 is removed to make each digit appear as a 4. Interestingly, the algorithm

failed to convert several digits into a 4 and instead converts them into other digits which are presumably more similar to that instance, despite the addition of the weighted term encouraging the latent representation to prefer the target class.

This type of failure is observed more in digits that are less similar to each other, such as from converting from the digits 3 to 2, as simply removing the lower curve of the digit may not always result in a centered "two" digit. This precludes the simple interpretation that we are able to attribute to the 9 to 4 task. This behaviour is not noted in our chest radiograph dataset as we are able to convert every image from the predicted class to the converse, which is presumably due to the smaller differences between chest X-rays with and without heart failure.

Similarly, the time taken to generate a visual rationale depends on the confidence of the classifier in its prediction, as the algorithm runs until the input has been altered sufficiently or a maximum number of steps (in our case 500) have been completed. In the case of converting digit 9s to 4s - we were able to generate 1000 visual rationales in 1 minute and 58 seconds.

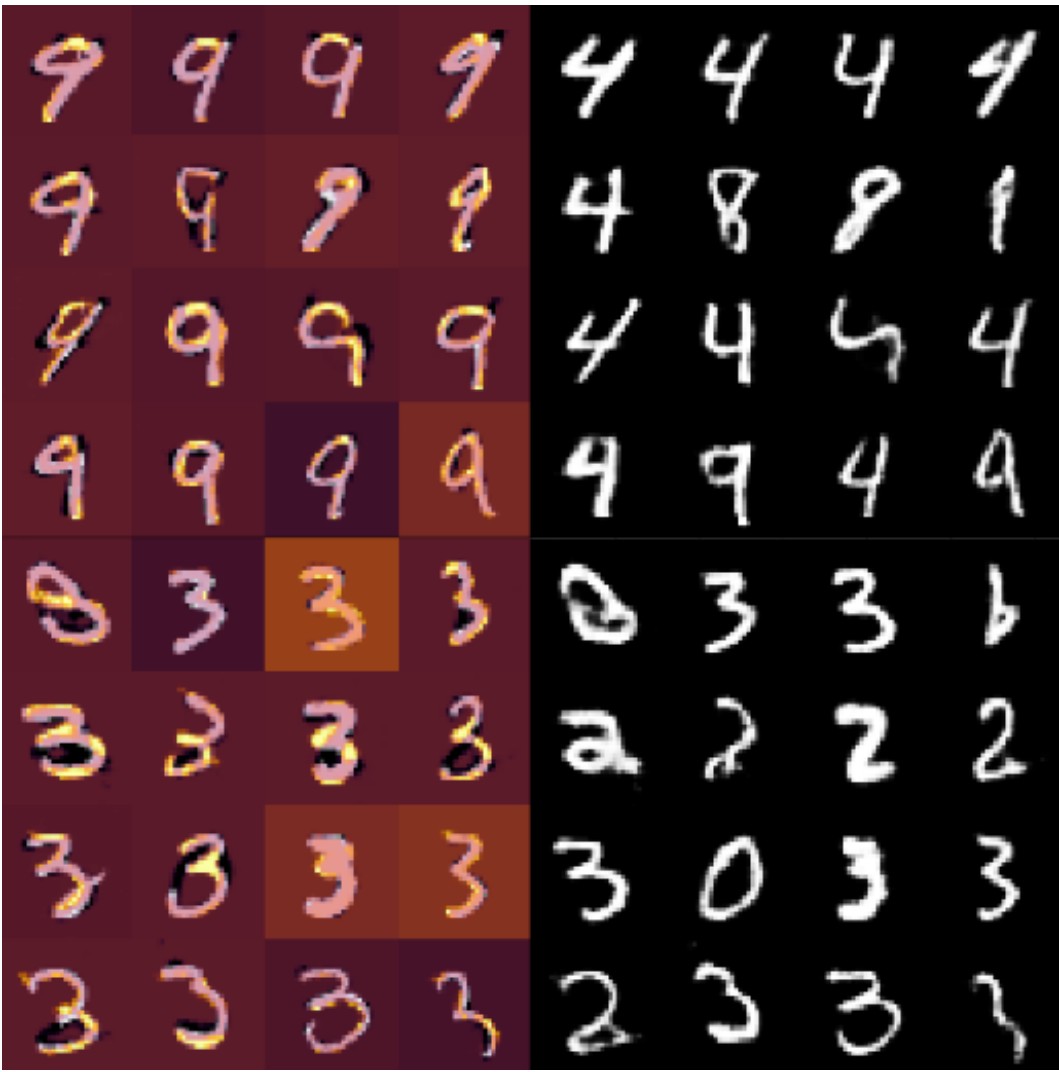

Figure 8: From left to right: original images with visual rationale overlaid, transformed digits

We compare this with the occlusion sensitivity and saliency map method demonstrated in Fig. 9.

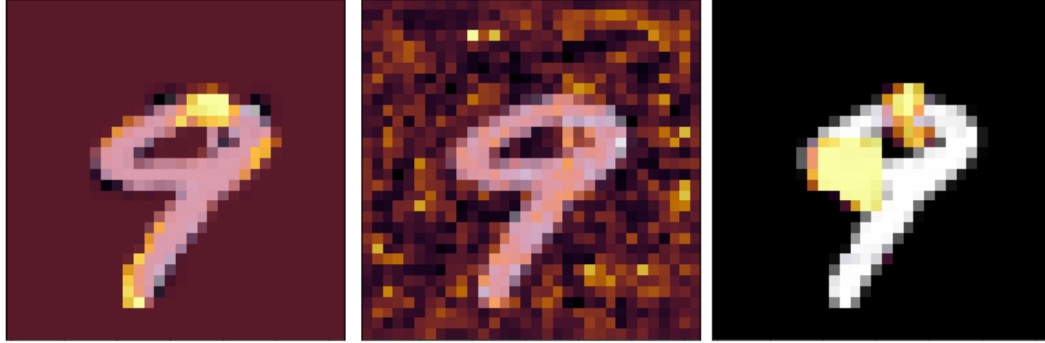

Figure 9: From left to right: visual rationale generated by our method, saliency map, occlusion sensitivity

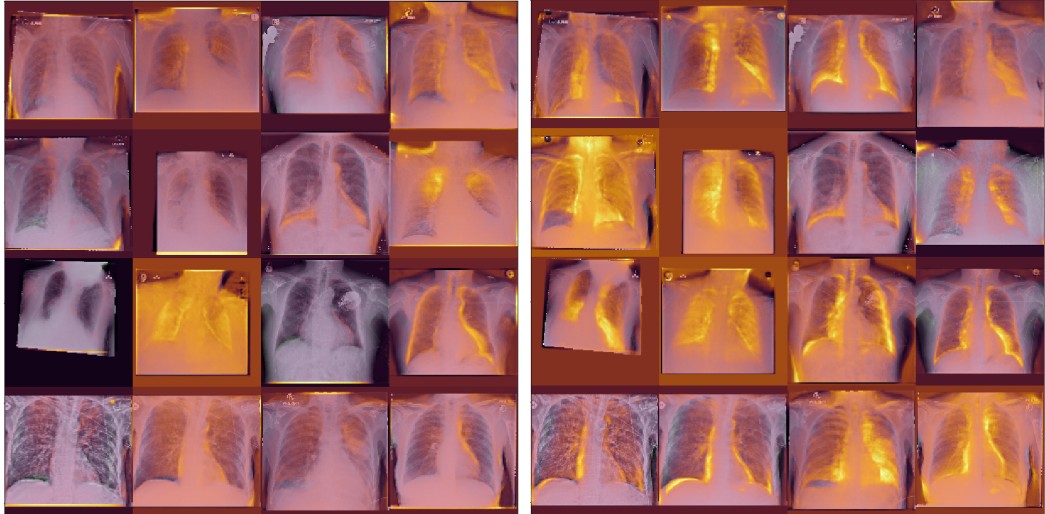

Figure 10: Left: Rationales from contaminated classifier. Right: rationales from normally trained classifier

Lastly, we contaminate our heart failure classifier as described in the methods section and compare visual rationales generated by the contaminated classifier with those generated previously. Fig 10. demonstrates images where both classifiers predict the presence of heart failure. The rationales from the contaminated classifier focus on small unique aspects of the image and largely do not correspond to our notion of what makes a chest radiograph more likely to represent heart failure, namely enlarged hearts and congested lung fields.

To demonstrate this we present 100 images classified as having a BNP level of over 100ng/L to two expert reviewers, equally split between a contaminated or a normally trained classifier. Each image and the associated visual rationale was presented to the reviewers who were blinded to the origin of the classifier. Reviewers were tasked in selecting features from a provided list which they felt corresponded with the visual rationales. Each reviewer rated each image twice. Aggregated results from experts are presented in Table 2. This clearly shows that the contaminated classifier indeed produces less interpretable visual rationales.

| Features | Reviewer A | | Reviewer B | |
|---|---|---|---|---|
| | Run 1 | Run 2 | Run 1 | Run 2 |
| Cardiomegaly | 35 | 34 | 44 | 46 |
| Effusion | 13 | 22 | 14 | 17 |
| Pacemaker | 1 | 1 | 1 | 1 |
| Airspace opacity | 9 | 6 | 3 | 9 |

(a) Correctly trained classifier. In Reviewer A's first round, 35 out of 50 visual rationales generated were identified as having features of cardiomegaly, and 13 had features of effusions. Each visual rationale can contain multiple features.

| Features | Reviewer A | | Reviewer B | |
|---|---|---|---|---|
| | Run 1 | Run 2 | Run 1 | Run 2 |
| Cardiomegaly | 22 | 19 | 18 | 22 |
| Effusion | 3 | 6 | 6 | 8 |
| Pacemaker | 3 | 4 | 3 | 3 |
| Airspace opacity | 4 | 3 | 3 | 4 |

(b) Contaminated classifier. In Reviewer A's first round, 22 out of 50 visual rationales generated were identified as having features of cardiomegaly, and 3 had features of effusions. Each visual rationale can contain multiple features.

Table 2: Number of radiographs with clinical features identified from visual rationales by expert reviewers.

## 4 DISCUSSION

We show in this work that using the generator of a GAN as the decoder of an autoencoder is viable and produces high quality autoencoders. The constraints of adversarial training force the generator to produce realistic radiographs for a given latent space, in this case a 100-dimensional space normally distributed around 0 with a standard deviation of 1.

This method bears resemblance to previous work done on inverting GANS done by Creswell & Bharath (2016), although we are not as concerned with recovering the exact latent representation but rather the ability to recreate images from our dataset. It is suggested in previous work in Kumar et al. (2017) that directly training a encoder to reverse the mapping learnt by the generator in a decoupled fashion does not yield good results as the encoder never sees any real images during training. By training upon the loss between the real input and generated output images we overcome this.

We further establish the utility of this encoder by using encoded latent representations to predict outcomes on unseen datasets, including one not from our institution. We achieve this without retraining our encoder on these unseen datasets, suggesting that the encoder has learnt useful features about chest radiographs in general.

Our primary contribution in this paper however is not the inversion of the generator but rather the ability to generate useful visual rationales. For each prediction of the model we generate a corresponding visual rationale with a target class different to the original prediction. We display some examples of the rationales this method produces and inspect these manually to check if these are similar to our understanding of how to interpret these images. The ability to autoencode inputs is essential to our rationale generation although we have not explored in-depth in this paper the effect of different autoencoding algorithms (for instance variational autoencoders) upon the quality of the generated rationales, as our initial experiments with variational and vanilla autoencoders were not able to reconstruct the level of detail required.

For chest radiographs, common signs of heart failure are an enlarged heart or congested lung fields, which appear as increased opacities in the parts of the image corresponding to the lungs. The rationales generated by the normally trained classifier in Fig 10 appear to be consistent with features described in the medical literature while the contaminated classifier is unable to generate these rationales.

We also demonstrate the generation of rationales with the MNIST dataset where the digit 9 is transformed into 4 while retaining the appearance of the original digit. We can see that the transformation generally removes the upper horizontal line of the 9 to convert this into a 4. Interestingly, some digits are not successfully converted. Even with different permutations of delta and gamma weights in Algorithm 2 some digits remain resistant to conversion. We hypothesize that this may be due to the relative difficulty of the chest radiograph dataset compared to MNIST — leading to the extreme confidence of the MNIST model that some digits are not the target class. This may cause vanishingly small gradients in the target class prediction, preventing gradient descent from achieving the target class.

We compare the visual rationale generated by our method to various other methods including integrated gradients, saliency maps, occlusion sensitivity as well as LIME in Fig. 6.

All of these methods share similarities in that they attempt to perturb the original image to examine the impact of changes in the image on the final prediction, thereby identifying the most salient elements. In the saliency map approach, each individual pixel is perturbed, while in the occlusion sensitivity method, squares of the image are perturbed. LIME changes individual superpixels in an image by changing all the pixels in a given superpixel to the average value. This approach fails on images where the superpixel classification is too coarse, or where the classification is not dependent on high resolution details within the superpixel. To paraphrase Sundararajan et al. (2017), attribution or explanation for humans relies upon counterfactual intuition — or altering the image to remove the cause of the predicted outcome. Model agnostic methods such as gradient based methods, while fulfilling the sensitivity and implementation invariance axioms, do not acknowledge the natural structure of the inputs. For instance, this often leads to noisy pixel-wise attribution as seen in Fig. 6. This does not fit well with our human intuition as for many images, large continuous objects dominate our perception and we often do not expect attributions to differ drastically between neighbouring pixels.

Fundamentally these other approaches suffer from their inability to perturb the image in a realistic fashion, whereas our approach perturbs the image's latent representation, enabling each perturbed image to look realistic as enforced by the GAN's constraints.

Under the manifold hypothesis, natural images lie on a low dimensional manifold embedded in pixel space. Our learned latent space serves as a approximate but useful coordinate system for the manifold of natural images. More specifically the image (pardon the pun) of the generator $G[\mathbb{R}^d]$ is approximately the set of 'natural images' (in this case radiographs) and small displacements in latent space around a point $z$ closely map into the tangent space of natural images around $G(z)$. Performing optimization in latent space is implicitly constraining the solutions to lie on the manifold of natural images, which is why our output images remain realistic while being modified under almost the same objective used for adversarial image generation.

Hence, our method differs from these previously described methods as it generates high resolution rationales by switching the predicted class of an input image while observing the constraints of the input structure. This can be targeted at particular classes, enabling us answer the question posed to our trained model — 'Why does this image represent Class A rather than Class B?'

There are obvious limitations in this paper in that we do not have a rigorous definition of what interpretability entails, as pointed out by Sundararajan et al. (2017). An intuitive understanding of the meaning of interpretability can be obtained from its colloquial usage — as when a teacher attempts to teach by example, an interpretation or explanation for each image helps the student to learn faster and generalize broadly without needing specific examples.

Future work could focus on the measurement of interpretability by judging how much data a second model requires when learning from the predictions and interpretations provided by another pre-trained model. Maximizing the interpretability of a model may be related to the ability of models to transfer information between each other, facilitating learning without resorting to the use of large scale datasets. Such an approach could help evaluate non-image based visual explanations such as sentences, as described in Hendricks et al. (2016).

Other technical limitations include the difficulty of training a GAN capable of generating realistic images larger than 128 by 128 pixels. This limits the performance of subsequent classifiers in identifying small features. This can be seen in the poor performance of our model in detecting nodules, a relatively small feature, compared to the baseline implementation in the NIH dataset.

In conclusion, we describe a method of semi-supervised learning and apply this to chest radiographs, using local data as well as recent datasets. We show that this method can be leveraged to generate visual rationales and demonstrate these qualitatively on chest radiographs as well as the well known MNIST set.

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
