# OpenReview forum: "Thinking like a machine — generating visual rationales through latent space optimization"
_ICLR.cc/2018/Conference — Reject_

### Official Review · AnonReviewer1 · 2017-11-24
**This paper models images with a latent code representation, and then tries to modify the latent code to minimize changes in image space, while changing the classification label. As the authors indicate, it lies in the space of algorithms looking to modify the image while changing the label (e.g. LIME etc).**

**Rating:** 4
**Confidence:** 3

**Review:**

* This paper models images with a latent code representation, and then tries to modify the latent code to minimize changes in image space, while changing the classification label. As the authors indicate, it lies in the space of algorithms looking to modify the image while changing the label (e.g. LIME etc).

* This is quite an interesting paper with a sensible goal. It seems like the method could be more informative than the other methods.  However, there are quite a number of problems, as explained below.

* The explanation of eqs 1 and 2 is quite poor. \alpha in (1) seems to be \gamma in Alg 1 (line 5). "L_target is a target objective which can be a negative class probability .." this assumes that the example is a positive class. Could we not also apply this to negative examples?

"or in the case of heart failure, predicted BNP level" -- this doesn't make sense to me -- surely it would be necessary to target an adjusted BNP level? Also specific details should be reserved until a general explanation of the problem has been made.

* The trade-off parameter \gamma is a "fiddle factor" -- how was this set for the lung image and MNIST examples? Were these values different?

* In typical ICLR style the authors use a deep network to learn the encoder and decoder networks. It would be v interesting (and provide a good baseline) to use a shallow network (i.e. PCA) instead, and elucidate what advantages the deep network brings.

* The example of 4/9 misclassification seems very specific. Does this method also work on say 2s and 3s? Why have you not reported results for these kinds of tasks?

* Fig 2: better to show each original and reconstructed image close by (e.g. above below or side-by-side).

The reconstructions show poor detail relative to the originals.  This loss of detail could be a limitation.

* A serious problem with the method is that we are asked to evaluate it in terms of images like Fig 4 or Fig 8. A serious study would involve domain experts and ascertain if Fig 4 conforms with what they are looking for.

* The references section is highly inadequate -- no venues of publication are given. If these are arXiv give the proper ref. Others are published in conferences etc, e.g. Goodfellow et al is in Advances in Neural Information Processing Systems 27, 2014.

* Overall: the paper contains an interesting idea, but given the deficiencies raised above I judge that it falls below the ICLR threshold.

* Text:

sec 2 para 4. "reconstruction loss on the validation set was similar to the reconstruction loss on the validation set." ??

* p 3 bottom -- give size of dataset

* p 5 AUC curve -> ROC curve

* p 6 Fig 4 use text over each image to better specify the details given in the caption.

---

> ### Author Response · Authors · 2017-12-26
> **Reply to reviewer 1**
>
> Thank you for your review and comments.
>
> 1) "The explanation of eqs 1 and 2 is quite poor. \alpha in (1) seems to be \gamma in Alg 1 (line 5). "L_target is a target objective which can be a negative class probability .." this assumes that the example is a positive class. Could we not also apply this to negative examples?"
>
> Thank you for pointing out the errors - textual details in Alg 1 and Eqs 1 and 2 have been fixed. This method can equally be applied to negative class, one need only flip the sign of L_target to achieve this.
>
> 2) "or in the case of heart failure, predicted BNP level -- this doesn't make sense to me -- surely it would be necessary to target an adjusted BNP level? Also specific details should be reserved until a general explanation of the problem has been made."
>
> We have removed the specific details at this stage of the paper.
>
> 3) "The trade-off parameter \gamma is a "fiddle factor" -- how was this set for the lung image and MNIST examples? Were these values different?"
>
> The trade-off parameter \gamma is indeed a ‘fiddle factor’ which was determined by the percentage of classes that were successfully switched while optimizing the latent space. As MNIST for instance is an easier problem than classifying heart failure, the classifier is more confident in predicting classes. The parameter gamma attempts to capture this by allowing more of the image to change in order to change the prediction of the classifier. In future work we hope to be able to derive a method of estimating gamma from the uncertainty of the predicted class probabilities but currently without an objective way of assessing these visual rationales we are unable to do so.
>
> 4) "In typical ICLR style the authors use a deep network to learn the encoder and decoder networks. It would be v interesting (and provide a good baseline) to use a shallow network (i.e. PCA) instead, and elucidate what advantages the deep network brings."
>
> As mentioned in the original paper, we did not test other methods of encoding and decoding images, for instance variational autoencoders or as suggested, shallower methods such as PCAs. However since the first draft of the paper, we have tried vanilla autoencoders as well as VAEs which fail to demonstrate the same ability to reconstruct images to the level of detail required - and we believe that PCA would run into similar obstacles.
>
> 5) "The example of 4/9 misclassification seems very specific. Does this method also work on say 2s and 3s? Why have you not reported results for these kinds of tasks?"
>
> This method also works for different number sets, including 2 and 3, however with differing rates of success. We have included a set of 3s to 2s in the updated version of our paper to illustrate this. As mentioned in the reply to Reviewer 3, this type of failure is observed more in digits that are less similar to each other, such as from converting from the digits 3 to 2, as simply removing the lower curve of the digit may not always result in a centered "two" digit. This precludes the simple interpretation that we are able to attribute to the 9 to 4 task.
>
> 6) "Fig 2: better to show each original and reconstructed image close by (e.g. above below or side-by-side). The reconstructions show poor detail relative to the originals.  This loss of detail could be a limitation."
>
> Figure 2 has been updated with your suggestion that the reconstructions be presented side by side for easier evaluation. You are correct in that the loss of detail could be a limitation - in fact we chose the training method we used (pretraining a GAN as the decoder part of an autoencoder) to preserve as much detail as possible (at the time of writing). The loss of detail means that our model is unable to explain predictions based on finer detail and we hope that future advances in generative learning will help overcome this.
>
> 7) "A serious problem with the method is that we are asked to evaluate it in terms of images like Fig 4 or Fig 8. A serious study would involve domain experts and ascertain if Fig 4 conforms with what they are looking for."
>
> We have included a blinded survey of domain experts in radiology in our revised paper to address the concern that readers may not be able to evaluate the images in Fig 4. This clearly demonstrates that the contaminated classifier produces visual rationales with fewer relevant features.
>
> 8) "The references section is highly inadequate -- no venues of publication are given. If these are arXiv give the proper ref. Others are published in conferences etc, e.g. Goodfellow et al is in Advances in Neural Information Processing Systems 27, 2014."
>
> Our references have been updated to include venues of publication as far as possible.

---

> > ### Comment · AnonReviewer1 · 2018-01-10
> > **There are still significant problems with this paper**
> >
> > I have read the responses to all reviewers, and the revised paper.
> >
> > I think there are still two significant problems with the paper
> >
> > 1) The paper still does not properly separate the underlying theoretical idea, and specific implementational details in section 2 (Methods).
> >
> > The key idea, that L_{target}(z) should depend on the *flipped* or *transformed* target is not articulated at all clearly. We have on p 3 only "L_{target}(z) is a target objective which can be a class probability or a regression target". This is inadequate.
> >
> > The basic idea of the paper should be properly specified first, then details like the "DCGAN ... with Scaled Exponential Linear Units"  (and a whole load more deep net alchemy) need to come later.
> >
> > Why does this matter? -- because if the idea is to be applicable to  other domains, all the domain-specific detail obliterates the key idea of the paper.
> >
> > 2) The authors say they have "included a blinded survey of domain  experts in radiology [...] to address the concerns that readers may not be able to evaluate the images in Fig 4."
> >
> > This is welcome, but Table 2 is COMPLETELY INCOMPREHENSIBLE. What are A1, A2, B1, B2? What are the 4 row labels? How do these relate to the difference images (as per Fig 5)? The authors need to EXPLAIN in the text what is going on in this table, and how this "clearly shows that the contaiminated classifier indeed produces less interpretable visual rationales".  I am also not sure that I really care about the contaminated classifier -- what I want to know is how the domain experts were able to use the difference image to aid their interpretations.
> >
> > I do note the positive scores of the other reviewers. I believe there is a good idea in this paper, but I still feel it is not explained properly, nor is the important domain expert evidence properly explained. To me it is still below threshold.

---

> > > ### Author Response · Authors · 2018-01-12
> > > **Explanation of table 2 and reply to comments**
> > >
> > > Thank you for your response to our comments - our reply to your concerns are as follows:
> > >
> > > "The authors say they have "included a blinded survey of domain experts in radiology [...] to address the concerns that readers may not be able to evaluate the images in Fig 4.""
> > >
> > > Domain experts (radiologists) were consulted throughout our project which formed the driving force for us to produce these interpretable visual rationales. These visual rationales are intended to help domain experts build confidence in the model by demonstrating that features identified by the model correlate with features in the medical literature.
> > >
> > > In order to show this, simply asking domain experts what features are being identified by the model is not useful - for instance the fact that Reviewer A identified that 35 out of 50 visual rationales were consistent with that of heart failure is not particularly informative without something to compare to.
> > >
> > > Hence, we demonstrate that when using a model that is incorrectly trained (i.e. one that does not split its training and test data and hence overfits), generated visual rationales show less useful features, and in fact spurious features not normally used in radiograph interpretation (e.g. pacemakers). The lack of features is likely to trigger suspicion in the end user that the model may in fact be incorrectly trained.
> > > 100 images with accompanying visual rationales were reviewed by domain experts. 50 images had their visual rationales generated by a correctly trained model and 50 by an incorrectly trained model. All images were predicted positive by their respective models. Two reviewers were sought (Reviewers A and B) who saw these 100 images twice in a randomized order, resulting in the columns A1, A2, B1 and B2.
> > >
> > > Hence, out of the 50 radiographs predicted positive for heart failure in the correctly trained model, reviewer A identified 35 with cardiomegaly in their first run and 34 in the second run through. Each reviewer rated each image twice to demonstrate the intra as well as inter observer differences, which allows comparison between the visual rationales produced by the correctly and incorrectly trained models. The results, while not statistically analysed, shows that the incorrectly trained model demonstrates less recognizable features - suggesting that this may be a useful tool for end users to help decide if the model can be relied upon.
> > >
> > > We are encouraged that you see the merit in our idea and hope that our explanation helps in the understanding of this table.
> > >
> > >
> > >
> > > "The paper still does not properly separate the underlying theoretical idea, and specific implementational details in section 2 (Methods)."
> > >
> > > Our paper is presented in a format that seeks reproducibility and hence follows a chronological development of each component. As you have pointed out, a drawback of this approach is that we do not properly separate the underlying theoretical idea from the specific implementation details, and domain specific details such as DCGANs are introduced prior to the key idea of the paper. We agree that more exposition on the underlying idea would be beneficial and further experiments could be conducted on different specific implementations in future work.

---

### Official Review · AnonReviewer2 · 2017-11-26
**Review of "Thinking like a machine — generating visual rationales through latent space optimization "**

**Rating:** 8
**Confidence:** 2

**Review:**

The main contribution of the paper is a method that provides visual explanations of classification decisions. The proposed method uses
 - a generator trained in a GAN setup
 - an autoencoder to obtain a latent space representation
 - a method inspired by adversarial sample generation to obtain a generated image from another class - which can then be compared to the original image (or rather the reconstruction of it).
The method is evaluated on a medical images dataset and some additional demonstration on MNIST is provided.


 - The paper proposes a (I believe) novel method to obtain visual explanations. The results are visually compelling although most results are shown on a medical dataset - which I feel is very hard for most readers to follow. The MNIST explanations help a lot.  It would be great if the authors could come up with an additional way to demonstrate their method to the non-medical reader.

 - The paper shows that the results are plausible using a neat trick. The authors train their system with the testdata included which leads to very different visualizations. It would be great if this analysis could be performed for MNIST as well.


From the related work, it would be nice to mention that generative models (p(x|c)) also often allow for explaining their decisions, e.g. the work by Lake and Tenenbaum on probabilistic program induction.
Also, there is the work by Hendricks et al on Generating Visual Explanations. This should probably also be referenced.

minor comments:
- some figures with just two parts are labeled "from left to right" - it would be better to just write left: ... right: ...
- figure 2: do these images correspond to each other? If yes, it would be good to show them pairwise.
- figure 5: please explain why the saliency map is relevant. This looks very noisy and non-interesting.

---

> ### Author Response · Authors · 2017-12-26
> **Reply to reviewer 2**
>
> Thank you for your review and comments. We were unaware of the work by Hendricks et al on Generating Visual Explanations and have sought to reference this in our discussion.
>
> In response to your comments:
>
> 1) "Some figures with just two parts are labeled "from left to right" - it would be better to just write left: ... right: …"
> 2) "Figure 2: do these images correspond to each other? If yes, it would be good to show them pairwise."
>
> We have rewritten our figure caption labels and also rearranged Figure 2 to demonstrate the original and reconstructed images pairwise for ease of comparison.
>
> 2) "Figure 5: please explain why the saliency map is relevant. This looks very noisy and non-interesting”
>
>  In Figure 5, the saliency map is indeed noisy and this serves to illustrate the deficiencies of the saliency map compared to the visual rationale generated using our method. We have added a statement in our paper to reflect this.

---

> > ### Comment · AnonReviewer2 · 2018-01-11
> > **Please explain Table 2**
> >
> > Dear authors,
> >
> > thanks for your changes.  I think overall the paper improved.
> >
> > The newly added Table 2 however is entirely ununderstandable.This definitely needs a better caption and possibly more description in the text.

---

> > > ### Author Response · Authors · 2018-01-12
> > > **Explanation of Table 2**
> > >
> > > Thank you for your comments - Reviewer 1 has also requested further explanation of Table 2 which we have detailed in our response to their comment below.

---

### Official Review · AnonReviewer3 · 2017-11-26
**Novel approach addressing important problems**

**Rating:** 7
**Confidence:** 4

**Review:**

The authors address two important issues: semi-supervised learning from relatively few labelled training examples in the presence of many unlabelled examples, and visual rationale generation: explaining the outputs of the classifiier by overlaing a visual rationale on the original image. This focus is mainly on medical image classification but the approach could potentially be useful in many more areas. The main idea is to train a GAN on the unlabeled examples to create a mapping from a lower-dimensional space in which the input features are approximately Gaussian, to the space of images, and then to train an encoder to map the original images into this space minimizing reconstruction error with the GAN weights fixed. The encoder is then used as a feature extractor for classification and regression of targets (e.g. heard disease). The visual rationales are generated by optimizing the encoded representation to simultaneously reconstruct an image close to the original and to minimize the probability of the target class. This gives an image that is similar to the original but with features that caused the classification of the disease removed. The resulting image can be subtracted from the original encoding to highlight problematic areas. The approach is evaluated on an in-house dataset and a public NIH dataset, demonstrating good performance, and illustrative visual rationales are also given for MNIST.

The idea in the paper is, to my knowledge, novel, and represents a good step toward the important task of generating interpretable visual rationales. There are a few limitations, e.g. the difficulty of evaluating the rationales, and the fact that the resolution is fixed to 128x128 (which means discarding many pixels collected via ionizing radiation), but these are readily acknowledged by the authors in the conclusion.

Comments:
1) There are a few details missing, like the batch sizes used for training (it is difficult to relate epochs to iterations without this). Also, the number of hidden units in the 2 layer MLP from para 5 in Sec 2.
2) It would be good to include PSNR/MSE figures for the reconstruction task (fig 2) to have an objective measure of error.
3) Sec 2 para 4: "the reconstruction loss on the validation set was similar to the reconstruction loss on the validation set" -- perhaps you could be a little more precise here. E.g. learning curves would be useful.
4) Sec 2 para 5: "paired with a BNP blood test that is correlated with heart failure" I suspect many readers of ICLR, like myself, will not be well versed in this test, correlation with HF, diagnostic capacity, etc., so a little further explanation would be helpful here. The term "correlated" is a bit too broad, and it is difficult for a non-expert to know exactly how correlated this is. It is also a little confusing that you begin this paragraph saying that you are doing a classification task, but then it seems like a regression task which may be postprocessed to give a classification. Anyway, a clearer explanation would be helpful. Also, if this test is diagnostic, why use X-rays for diagnosis in the first place?
5) I would have liked to have seen some indicative times on how long the optimization takes to generate a visual rationale, as this would have practical implications.
6) Sec 2 para 7: "L_target is a target objective which can be a negative class probability or in the case of heart failure, predicted BNP level" -- for predicted BNP level, are you treating this as a probability and using cross entropy here, or
mean squared error?
7) As always, it would be illustrative if you could include some examples of failure cases, which would be helpful both in suggesting ways of improving the proposed technique, and in providing insight into where it may fail in practical situations.

---

> ### Author Response · Authors · 2017-12-26
> **Reply to Reviewer 3**
>
> Thank you for the comments and your review.  Your description of our process is accurate. We have addressed each of your comments.
>
> 1) “There are a few details missing, like the batch sizes used for training (is it difficult to relate epochs to iterations without this). Also, the number of hidden units in the 2 layer MLP from para 5 in sec 2”
>
> In this updated version, we have included batch sizes and the number of hidden units in our methods section.
>
> 2) “It would be good to include PSNR/MSE figures for the reconstruction task (fig 2) to have an objective measure of error”
> 3) “Sec 2 para 4: the reconstruction loss on the validation set was similar to the reconstruction loss on the validation set -- perhaps you could be a little more precise here. E.g. learning curves would be useful."
>
>
> We have included additional figures showing the Laplacian loss functions for training and testing sets as well as corresponding MSE figures. This illustrates our point that when the decoder is fixed, overfitting for the autoencoder is not observed.
>
> 4) "Sec 2 para 5: paired with a BNP blood test that is correlated with heart failure" I suspect many readers of ICLR, like myself, will not be well versed in this test, correlation with HF, diagnostic capacity, etc., so a little further explanation would be helpful here. The term "correlated" is a bit too broad, and it is difficult for a non-expert to know exactly how correlated this is. It is also a little confusing that you begin this paragraph saying that you are doing a classification task, but then it seems like a regression task which may be postprocessed to give a classification. Anyway, a clearer explanation would be helpful. Also, if this test is diagnostic, why use X-rays for diagnosis in the first place?"
>
> We have updated the BNP section to clarifying some important points that you've brought up. Even in the medical literature, the diagnosis of heart failure is not well defined and usually relies on a mix of patient symptoms, BNP results, and radiology. Whilst not readily available in every hospital services, BNP serves as an objective measure to diagnose heart failure and is being increasingly used by clinicians. Hence these are useful to predict as they represent an objective label for the chest X-ray, whereas current deep learning methods tend to utilize radiologist reports of the X-ray image which can often omit diagnoses that were deemed irrelevant by the radiologist.
>
> BNP levels are continuous and hence we train our network as a regression task, however we evaluate this using AUC as clinicians are often interested specifically if BNP levels are over a laboratory-defined threshold, and AUC is often the metric used in the medical literature for comparing the diagnostic capacities of different tests. Lastly, BNP tests are not available in all laboratories and may take a while to return while chest X-ray images are easily available although tricky to interpret, even for medical doctors, as outlined in Kennedy et al (2011).
>
> 5) "I would have liked to have seen some indicative times on how long the optimization takes to generate a visual rationale, as this would have practical implications."
>
> Indicative times have been added in our results section as well. Times may vary depending on the confidence of the classifier as inputs that do not lie close to the target class may take more steps to convert or in fact may fail to convert if the maximum number of steps have been completed.
>
> 6) "Sec 2 para 7: L_target is a target objective which can be a negative class probability or in the case of heart failure, predicted BNP level -- for predicted BNP level, are you treating this as a probability and using cross entropy here, or mean squared error?"
>
> For predicted BNP level we are using mean squared error - as the network was trained on the regression task of predicting the BNP level
>
>
> 7) "As always, it would be illustrative if you could include some examples of failure cases, which would be helpful both in suggesting ways of improving the proposed technique, and in providing insight into where it may fail in practical situations."
>
> We have included (also based on the suggestions of Reviewer 1) other examples on MNIST - in particular changing the predicted class from 3 to 2. This is a significantly harder task as most digits are centered in the MNIST dataset and hence we cannot simply remove the bottom curve of the 3 to convert it to a 2, as we can with a 9 to a 4. This generates several failure cases where the algorithm instead converts the 3 into something else, or fails to convert it at all.

---

> > ### Comment · AnonReviewer3 · 2018-01-12
> > **Table 2 caption**
> >
> > Thank you for updating the paper. I am satisfied with the changes.
> >
> > However, and as noted by the other reviewers, the description of the newly added Table 2 in the paper is very unclear. I needed to read your response to reviewer 1 to understand what was being presented. Please update the paper with a comprehensible description (ideally in the caption to Table 2).

---

> > > ### Author Response · Authors · 2018-01-16
> > > **Revised Table 2**
> > >
> > > Thank you for your reply - as per your request Table 2 has been updated to include some of our response to Reviewer 1 as a caption to help understand the table's contents.

---

### Author Response · Authors · 2017-12-26
**Summary of changes made in second revision of paper**

We would like to thank all the reviewers for their comments and contributions. The paper has been modified with several suggestions from all reviewers included, namely:

* Added domain expert ratings for visual rationales produced by correctly and incorrectly trained algorithms
* Added figure for autoencoder training v.s. validation loss functions
* Symbols fixed in equations and algorithms
* References updated with correct publication venues
* Textual and spelling errors corrected
* Added section explaining the choice of BNP as the label for chest X-rays and the real world applications of this
* Edited figure 2 so that original and reconstructed images are displayed pairwise
* Add indicative times in Results (Sec 3)
* Added ChestX-ray8 dataset size
* Added batch sizes and hidden units for classification MLP
* Added references to ‘Generating Visual Explanations’ in Discussion
* Edited caption and accompanying text for Figure 5 to explain why the saliency map is more relevant
* Included additional MNIST examples as well as failure cases.

---

### Decision · Program_Chairs · 2018-01-29
**ICLR 2018 Conference Acceptance Decision**

**Decision:**

Reject

**Comment:**

The paper proposes a semi-supervised method to make deep learning more interpretable and at the same time be accurate on small datasets. The main idea is to learn dense representations from unlabelled data and then use those for building classifiers on small datasets as well as generate visual explanations. The idea is interesting, however, as one reviewer points out the presentation is poor. For instance, Table 2 is not understandable. Given the high standards of ICLR this cannot be ignored especially given the fact that the authors had the benefit of updating the paper which is a luxury for conference submissions.